# Genome-Wide Characterization and Analysis of bHLH Transcription Factors Related to Anthocyanin Biosynthesis in *Cinnamomum camphora* (‘Gantong 1’)

**DOI:** 10.3390/ijms24043498

**Published:** 2023-02-09

**Authors:** Xue Gong, Tengfei Shen, Xiuqi Li, Hanbin Lin, Caihui Chen, Huihu Li, Zhaoxiang Wu, Qiaoli Liu, Meng Xu, Bo Zhang, Yongda Zhong

**Affiliations:** 1College of Forestry, Jiangxi Agricultural University, Nanchang 330096, China; 2Key Laboratory of Horticultural Plant Genetics and Improvement of Jiangxi Province, Institute of Biological Resources, Jiangxi Academy of Sciences, Nanchang 330096, China; 3Co-Innovation Center for Sustainable Forestry in Southern China, Key Laboratory of Forest Genetics and Biotechnology of Ministry of Education, Nanjing Forestry University, Nanjing 210037, China; 4Umeå Plant Science Centre, Department of Forest Genetics and Plant Physiology, Swedish University of Agricultural Sciences, 901 93 Umeå, Sweden

**Keywords:** anthocyanin, *Cinnamomum camphora*, basic helix–loop–helix (bHLH) transcription factors, collinearity analysis, RNA sequencing

## Abstract

*Cinnamomum camphora* is one of the most commonly used tree species in landscaping. Improving its ornamental traits, particularly bark and leaf colors, is one of the key breeding goals. The basic helix–loop–helix (bHLH) transcription factors (TFs) are crucial in controlling anthocyanin biosynthesis in many plants. However, their role in *C. camphora* remains largely unknown. In this study, we identified 150 bHLH TFs (*CcbHLHs*) using natural mutant *C. camphora* ‘Gantong 1’, which has unusual bark and leaf colors. Phylogenetic analysis revealed that 150 CcbHLHs were divided into 26 subfamilies which shared similar gene structures and conserved motifs. According to the protein homology analysis, we identified four candidate CcbHLHs that were highly conserved compared to the TT8 protein in *A. thaliana*. These TFs are potentially involved in anthocyanin biosynthesis in *C. camphora*. RNA-seq analysis revealed specific expression patterns of *CcbHLHs* in different tissue types. Furthermore, we verified expression patterns of seven *CcbHLHs* (*CcbHLH001*, *CcbHLH015*, *CcbHLH017*, *CcbHLH022*, *CcbHLH101*, *CcbHLH118*, and *CcbHLH134*) in various tissue types at different growth stages using qRT-PCR. This study opens a new avenue for subsequent research on anthocyanin biosynthesis regulated by *CcbHLH* TFs in *C. camphora*.

## 1. Introduction

*Cinnamomum camphora* is an evergreen tree species with a large canopy and dense shade all year round. *C. camphora* is one of the most commonly used tree species in landscaping, with high economic, ornamental, and ecological value [1]. *C. camphora* is a native landscape tree in southern China, and its ornamental traits, such as bark and leaf colors, are important selection targets. Many studies have shown that the color of the bark and leaves of angiosperms depends on the proportion and distribution of four pigments: chlorophylls, flavonoids, carotenoids, and betaine [2]. Our previous studies on the transcriptome and metabolome of *C. camphora* demonstrated that red bark results from an enrichment of anthocyanins [3]. These pigments are widely distributed in the leaves, flowers, fruits, and vegetative tissues of vascular plants. Plant anthocyanins also help them to cope with biotic and abiotic stresses. For example, increased anthocyanin accumulation in grape leaves improves their resistance to cold temperatures [4]. Compared with the sensitivity of other orange varieties, the anthocyanin-rich blood orange ‘Tarocco’ is less susceptible to the necrotizing fungus (*Penicillium digitatum*) that causes green mold [5].

The pathways of anthocyanin biosynthesis in plants have been progressively elucidated in recent years. Anthocyanin biosynthesis starts from phenylalanine. After a series of enzymatic catalytic steps, the final products are transported to vacuoles and other parts for storage [6]. The MBW (MYB, bHLH, and WD40) transcription complex comprises the major transcription factors (TFs) regulating anthocyanin biosynthesis in most plants [7]. The MBW complex directly controls the transport and modification of structural genes for anthocyanin biosynthesis [8,9]. We previously reported that many structural genes related to anthocyanin synthesis, such as *PAL*, *4CL*, *CHS*, *F3H*, *F3′H*, *DFR*, *LAR*, *ANR*, *UGTs*, and *OMT*, have been verified by qRT-PCR, and their expression levels in ‘Gantong 1’ were significantly higher than in their half-sib progeny [3]. However, the TFs regulating anthocyanin synthesis have not been studied in detail.

The bHLH TF family, named after its highly conservative basic/helix–loop–helix domain, is widely represented in plants [10,11]. The bHLH domain comprises 50–60 amino acids divided into the *basic* and HLH sub-domains. The *basic* domain contains 13–18 amino acids at the N-terminal, which binds to the E-box (5′-CANNTG-3) [12]. The C-terminal HLH domain is mainly involved in forming the homo- or heterodimers [13], which determines other binding sites of the target genes’ promoter. The C-terminal HLH domain contributes to the specificity of transcriptional regulation in eukaryotes [14]. So far, various numbers of bHLH TFs have been identified in different species, including 162 in *Arabidopsis thaliana*, 183 in *Populus trichocarpa*, 197 in *Pyrus bretschneideri*, 115 in *Vitis davidii*, 91 in *Liriodendron chinense*, 134 in *Camellia sinensis*, 118 in *Ficus carica*, and 95 in *Gardenia jasminoides*, respectively [15,16,17,18,19,20,21,22]. Most bHLH TF family studies focus on anthocyanin biosynthesis. Song et al. identified a total of 16 candidate genes related to anthocyanin synthesis among 118 bHLH transcription factors in *Ficus carica*, and performed transient expression verification of *FcbHLH42*, confirming that this gene promotes anthocyanin synthesis and accumulation [21]. For consistency, Zhao et al identified 102 bHLH genes in the *Juglans regia* genome, 4 of which were involved in anthocyanin synthesis [23]. bHLH TFs that regulate the biosynthesis of anthocyanins and flavonoids mainly belong to the Ⅲf subfamily, such as TT8 (TRANSPARENT TESTA 8), GL3 (GLABRA3), and EGL3 (ENHANCER OF GLABRA3) in *A. thaliana* [24]. Some bHLH TFs regulate the transcription of anthocyanin biosynthesis genes. For instance, TT8 mainly regulates the expression of the dihydroflavonol 4-reductase (*DFR*) gene in seedlings and pods and participates in the formation of the MBW complex [25]. The transcription factor DhbHLH1 regulates the co-expression of *DhDFR* and *DhANS* in the petals of *Dendrobium*, thereby modulating anthocyanin production [26]. *TfbHLH1* plays a significant role in the regulation of anthocyanin biosynthesis in tulip tepals [27]. Arlotta et al. suggested that *PgbHLH* may be involved in the determination of differences in flavonoid composition in both flowers and fruits of the pomegranate varieties Wonderful and Valenciana [28].

The leaves and bark of normal *C. camphora* are usually dark green and dark brown, respectively. In contrast, the bark and leaves of the natural mutant *C. camphora* ‘Gantong 1’ are red to dark red and orange-red to yellow-green, respectively. Hence, the characterization of bark and leaf coloration of ‘Gantong 1’ would significantly contribute to our understanding of the coloring mechanism of *C. camphora*. The recently released whole genome sequence of *C. camphora* is a valuable tool to study the bHLH TF family of *C. camphora* in detail [29]. In this study, we analyzed 150 bHLH TFs in *C. camphora* at the genomic level, revealed their potential roles in the regulation of anthocyanin biosynthesis, and provided valuable insights for genetic improvement and molecular breeding focused on the coloration of the ornamental tree *C. camphora*.

## 2. Results

### 2.1. Identification and Physiochemical Characteristics of CcbHLHs

In total, 173 *bHLH* TF family members were identified by the Hidden Markov Model (PF00010) in the *C. camphora* genome. Subsequently, we used the NCBI Batch CD-Search database and SMART to determine the presence of the conserved bHLH domain and remove the redundant sequences without the bHLH domain. Finally, 150 members of the *CcbHLH* TF family were identified from the whole genome protein file of *C. camphora*. These genes were named *CcbHLH*1–150 based on their chromosomal position (Appendix A). The 150 *CcbHLHs* were randomly distributed on 12 *C. camphora* chromosomes, with 7–22 *CcbHLHs* on each chromosome. Chromosome 4 had the most *CcbHLHs* (22), followed by chromosome 2 (20) and chromosome 1 (17). Chromosomes 8 and 12 had 7 *CcbHLHs* each. A total of 143 *bHLH* TFs were identified in the *C. kanehirae* genome, which was 7 fewer than the number of *CcbHLHs* in the genome of *C. camphora*.

Then, the physicochemical properties of CcbHLH proteins were computed by using Protparam EXPASY online software. CcbHLH proteins ranged from 86 (CcbHLH073) to 1216 amino acids (CcbHLH091) in length. The molecular weight of CcbHLH proteins ranged from 10.04 kD (CcbHLH110) to 136.68 kD (CcbHLH091). The hydrophilicity of CcbHLH proteins ranged from −1.016 (CcbHLH066) to −0.086 (CcbHLH021), indicating that all CcbHLH proteins are hydrophilic. The theoretical isoelectric points of CcbHLH proteins ranged between 4.49 (CcbHLH032) and 11.98 (CcbHLH086). The instability index of CcbHLH proteins ranged from 37.35 (CcbHLH046) to 86.82 (CcbHLH047), and only CcbHLH008, CcbHLH046, CcbHLH109, and CcbHLH148 proteins had instability indices less than 40, indicating that most CcbHLH proteins may be unstable (Appendix A).

### 2.2. Phylogenetic Analysis of CcbHLHs

The recently reported numbers of bHLH protein subpopulations in different plant species ranged between 15 and 32 [16,30,31]. The reported bHLH TF family members of *A. thaliana* (162) were downloaded from the TAIR databases [15]. After multiple alignments of the protein sequences of *C. camphora* and *A. thaliana* using MAFFT version 7, a rootless phylogenetic tree was constructed based on 162 AtbHLHs (Figure 1 and Appendix A). We performed cluster analysis of CcbHLHs based on the classification method developed for *A. thaliana* [15,31,32] and made appropriate adjustments. The phylogenetic tree revealed that bHLHs from the two species could be separated into 28 clades. The members that did not belong to any subfamilies of *A. thaliana* (CcbHLH020, CcbHLH086, CcbHLH097, CcbHLH129, CcbHLH136, and CcbHLH149) were classified as “Orphans”. Twenty-eight clades coalesced into 27 subfamilies and 1 “Orphan” subfamily. CcbHLHs were distributed in 26 of these subfamilies, and 2 subfamilies, VI and XV, belonged to the specific family of *A. thaliana*. The number of members in different subfamilies varied greatly. Among them, Ib was the largest subfamily, with 21 members, followed by subfamilies XII and X, with 16 and 12 members, respectively. The smallest subfamilies, Va, XIII, IVb, and XIV, each had two CcbHLH members. The classification result of the *C. camphora* bHLH TF family based on the phylogenetic tree illustrates the evolutionary relationship of CcbHLHs.

### 2.3. Analysis of Structure and Conserved Motifs of CcbHLHs

Analysis of gene structures and conserved motifs can help elucidate *CcbHLH* functions (Appendix A). Twenty conserved motifs in total were predicted by online MEME software, and the members of the same subgroup tended to have the same or comparable motifs. Motif 1 was the most frequently occurring among all *CcbHLH* members, being present in more than 96% of *CcbHLH* TFs. Only *CcbHLH066*, *CcbHLH073*, *CcbHLH096*, *CcbHLH119*, and *CcbHLH148* did not contain Motif 1. Motif 2 was the second-most abundant, being present in more than 86% of *CcbHLH* TFs. Some motifs only appeared in specific subfamilies: for instance, motif 10 was only noted in subfamily XII, motif 17 was only found in subfamily Ib, and motif 14 was only found in subfamily VIIIb. Gene structure analysis indicated that the *CcbHLH* TF family members have a wide range of exon numbers as well as gene structural varieties. The number of exons in *CcbHLHs* ranged from 1 to 17, with 17 *CcbHLHs* having only 1 exon, whereas 4 *CcbHLHs* had more than 10 exons (Appendix A). The *CcbHLH* TFs were highly conserved within the same subfamily and had a broadly similar exon/intron structure.

### 2.4. Chromosomal Localization and Collinearity Analysis of CcbHLHs

Unknown genes can be quickly understood, located, and cloned by comparisons with genes and gene structures from well-characterized species [33]. The positions and sequences of genes on homologous chromosomes are similar between and within species, so a large number of collinear regions revealed by the gene collinearity analysis can be regarded as direct evidence of the whole-genome duplication [34]. Large chromosome repeats, tandem repeats, and transposition events are the key means of gene family amplification [35]. The results of intraspecies genome localization analysis showed that seven *CcbHLHs* were clustered into two tandem duplication regions (*CcbHLH067*/*CcbHLH068*/*CcbHLH069* and *CcbHLH125*/*CcbHLH126*/*CcbHLH127*/*CcbHLH128*) on *C. camphora* chromosomes 4 and 10, respectively. Among the 150 *CcbHLHs*, there were 68 pairs of segmentally duplicated genes (Figure 2). To understand whether *CcbHLHs* were subjected to natural selection during evolution, *Ka/Ks* analysis was performed on tandem-duplicated and fragmentally duplicated genes. The gene was considered to have undergone purification selection if *Ka*/*Ks* < 1 [36]. For all duplicated *CcbHLHs*, *Ka*/*Ks* values were below 1, indicating that *C. camphora* eliminated harmful mutations through purifying selection during evolution (Appendix A).

We also performed collinearity analysis on *C. camphora* and two other representative model plants, *A. thaliana* and *P. trichocarpa*. In total, 76 and 203 orthologous gene pairs were identified between *C. camphora* and *A. thaliana* and between *C. camphora* and *P. trichocarpa*, respectively (Figure 3), indicating a closer homologous evolutionary relationship of the *C. camphora* bHLH TF family with that of *P. trichocarpa* than with that of *A. thaliana*. There were 60 CcbHLHs that had no collinear gene pairs with the bHLH TF of *A. thaliana* and *P. trichocarpa*, indicating that these TFs were of different origin from *A. thaliana* and *P. trichocarpa*. Further experiments will be needed to characterize the functions of these *CcbHLHs*. Subsequently, we conducted collinear analysis between *C. camphora* and the related species *Cinnamomum kanehirae*, and found 256 pairs of homologous genes, among which 16 *CcbHLH* TFs were of a different origin than that of the *C. kanehirae* genes. It is possible that these 16 TFs have specific functions in *C. camphora* (Appendix A). In addition, the *Ka/Ks* values of the directly homologous *CcbHLHs* of *C. camphora*, *A. thaliana,* and *P. trichocarpa* were all less than 1, indicating a strong effect of purifying selection on the *CcbHLH* gene family (Appendix A). The *Ka*/*Ks* values of eight *CcbHLH* directly homologous genes of *C. camphora* and *C. kanehirae* were greater than 1, indicating their positive selection (Appendix A).

### 2.5. Predicted Protein–Protein Interaction Network of CcbHLHs

The bHLH TF family members typically function by forming homo- or heterodimers with other proteins, which is indispensable for their binding to the promoters of target genes. We used STRING to predict protein interaction networks based on CcbHLH direct homologs in *A. thaliana*. The results showed that most CcbHLHs interacted with more than 1 CcbHLH protein, and 19 CcbHLHs interacted with more than 10 CcbHLHs (Appendix A). CcbHLH001, CcbHLH022, CcbHLH118, and CcbHLH134 (TT8 in *A. thaliana*) cooperate with TT1, PAP1 and TTG1 to regulate biosynthesis of proanthocyanidins and anthocyanidins by modulating the expression of the *DFR* gene (Figure 4 and Appendix A) [18]. These results suggest that CcbHLH001, CcbHLH022, CcbHLH118, and CcbHLH134 may participate in anthocyanin biosynthesis.

### 2.6. Expression Levels of CcbHLHs in Different Tissues

To reveal the expression pattern of *CcbHLHs*, we performed transcriptome sequencing analysis of different *C. camphora* (‘Gantong 1’) tissues, including the stem, fruit, root, xylem, leaf, flower, and phloem. A hierarchical clustering analysis found significant differences in *CcbHLHs* expression patterns (Figure 5). In particular, 58, 64, 41, 55, 52, 50, and 59 *CcbHLHs* with relatively high expression levels (FPKM > 10) were observed in the phloem, flower, leaf, xylem, root, fruit, and stem tissues, respectively. Among them, 22 *CcbHLHs* (*CcbHLH002*, *CcbHLH006*, *CcbHLH009*, *CcbHLH011*, *CcbHLH022*, *CcbHLH034*, *CcbHLH041*, *CcbHLH045*, *CcbHLH051*, *CcbHLH071*, *CcbHLH080*, *CcbHLH090*, *CcbHLH105*, *CcbHLH117*, *CcbHLH118*, *CcbHLH120*, *CcbHLH131*, *CcbHLH134*, *CcbHLH137*, *CcbHLH145*, *CcbHLH148*, and *CcbHLH149*) showed high expression levels in all tissues (FPKM > 10). *CcbHLH042* showed the highest transcript accumulation in the phloem and leaf (FPKM > 150). Further, *CcbHLH005*, *CcbHLH097*, *CcbHLH120*, *CcbHLH148*, and *CcbHLH149* showed the highest expression level in the flower, xylem, root, fruit, and stem (FPKM > 147), respectively. However, 17 *CcbHLHs* (*CcbHLH003*, *CcbHLH035*, *CcbHLH050*, *CcbHLH054*, *CcbHLH062*, *CcbHLH066*, *CcbHLH067*, *CcbHLH077*, *CcbHLH078*, *CcbHLH092*, *CcbHLH098*, *CcbHLH104*, *CcbHLH113*, *CcbHLH122*, *CcbHLH123*, *CcbHLH127*, and *CcbHLH143*) had very low or no expression in all seven tissues tested (FPKM < 0.5).

### 2.7. Analysis of Expression Levels of Candidate TFs for Anthocyanin Biosynthesis

To further investigate the roles of *CcbHLH* TFs in anthocyanin biosynthesis in *C. camphora*, we characterized the expression levels of seven *CcbHLHs* in various tissue types at different stages by qRT-PCR. Three *CcbHLHs* were identified previously (*CcbHLH015*, *CcbHLH017*, and *CcbHLH101*) and four were revealed in this study (*CcbHLH001*, *CcbHLH022*, *CcbHLH118*, and *CcbHLH134*) (Figure 6). We compared the expression patterns of these TFs in the mutant ‘Gantong 1’ variety to those of the ‘Gantong 1’ half-sib progenies which were used as a genetic background control. In the ‘Gantong 1’ leaves, transcriptional levels of three *CcHLHs* (*CcbHLH001*, *CcbHLH015*, and *CcbHLH017*) peaked in April and May, while expression of *CcbHLH022* and *CcbHLH134* peaked in May and August. The expression of *CcbHLH101* reached its peak in July and August. However, the expression levels of *CcbHLH118* showed one peaks in April (Figure 6A). Furthermore, we observed significant differences in expression levels of six *CcbHLHs* in the ‘Gantong 1’ leaves compared to those in control plants. Subsequently, we analyzed the expression pattern of CcbHLH in bark, and found that the expression levels of *CcbHLH001* and *CcbHLH101* peaked in April and July, whereas expression of *CcbHLH022* and *CcbHLH134* peaked in January and August. *CcbHLH118* peaked in January and April, while *CcbHLH015* only peaked in April (Figure 6B). Interestingly, significant differences were found for some genes that were exclusively higher in ‘Gantong 1’ than in the control plants. Taken together, these results suggest potential roles of the characterized *CcbHLHs* in leaf and bark development.

## 3. Discussion

*C. camphora*, one of the essential landscaping species of the world, is widely used in street-side greening and to create shade. The colorful leaves, bark, and other ornamental traits are important breeding goals for *C. camphora*. bHLH TFs are the essential TFs regulating anthocyanin biosynthesis. The function of some *bHLH* TF family members in anthocyanin biosynthesis has been previously demonstrated in species such as *Vitis davidii*, *Ficus carica* L., and *Juglans regia* L. [18,21,23]. We previously reported that *bHLH* TFs are involved in anthocyanin synthesis in *C. camphora* (‘Gantong 1’) [3]. However, the roles of *bHLHs* in regulating the colors of *C. camphora* tissues remained unclear. The elucidation of genomic characteristics of *C. camphora* bHLH TFs could help understand the development of this ornamental trait.

In this study, 150 *CcbHLH* TF family members were identified in *C. camphora*, which was a similar number of bHLH TFs compared to that in *Panax ginseng* (169) and tomato (159) [14,37]. The number of TFs revealed in *C. camphora* was larger than that in *Liriodendron chinense* (91) [19], *Prunus avium* L. (66) [38], and *Juglans regia* L. (102) [23], but smaller than that in *Helianthus annuus* L. (183) [39] and *Pyrus bretschneideri* (197) [17]. These differences might result from differences in gene/genome duplication events during evolution. A total of 143 *bHLH* transcription factors were identified in *C. kanehirae* of the same genus as *C. camphora*, which was basically similar to *C. camphora*, indicating the same evolutionary relationship within the same genus. *C. camphora* is rich in essential oils, which contain terpenoids. The leaf essential oil of *C. camphora* is considered a contributor to the beneficial properties of this plant [40]. Terpene synthase (TPS) is a critical enzyme in terpene synthesis. MYC, a bHLH family transcription factor, regulates the expression of terpenoid biosynthetic genes in various plants. Hong et al. demonstrated that MYC2 in *A. thaliana* directly binds to the promoters of *TPS21* and *TPS11* and activates their expression, thereby promoting biosynthesis of (E)-β-caryophyllene and other terpenoids [41]. Therefore, *C. camphora* may contain more *CcbHLH* family members than other species to regulate *TPS* gene expression.

The phylogenetic tree results showed that 150 CcbHLH TFs were divided into 26 subfamilies, which is similar to the number of 26 reported for *Andrographis paniculata* [42], 24 in *Panax ginseng* [37], 25 in *Osmanthus fragrans* [43], and 25 in *A. thaliana* [32]. However, compared to *Arabidopsis*, the minor subfamily VI was not found in *C. camphora*, which may be attributed to the loss of genes during evolution. We then adopted the Pires’s classification method to analyze *CcbHLHs* that did not match Heim’s classification in *C. camphora*. These members then fell into subfamilies XIII, XIV, and Orphan. The numbers of CcbHLH TFs in subfamilies Ib, X, and XII were the largest, broadly in agreement with the pattern of each corresponding subfamily in *A. thaliana*. Subfamilies Va (CcbHLH074, CcbHLH140), XIII (CcbHLH119, CcbHLH135), IVb (CcbHLH038, CcbHLH117), and XIV (CcbHLH148, CcbHLH143) had two members each, suggesting a relatively slow evolution rate. So far, the biological functions of CcbHLHs still remain unclear. However, according to this study and previous studies on other plant species, we were able to narrow down the number of candidate *CcbHLH* TFs potentially involved in anthocyanin biosynthesis. For instance, CcbHH001, CcbHH022, CcbHH118, and CcbHH134 belonged to the subfamily IIIf, in which *A. thaliana* (AtbHLH042) was found, which is known to be involved in anthocyanin synthesis [25]. In addition, FcBHLH42 in *Ficus carica* L. [21], FabHLH29 in strawberry [31], FhGL3L and FhTT8L in *Freesia hybrida* [44], CmbHLH2 in *Chrysanthemum morifolium* R. [45], and VdbHLH037 in *Vitis davidii* [18] belonged to subfamily IIIf. These genes have also been proven to be involved in anthocyanin and proanthocyanin biosynthesis.

Further analysis of *CcbHLH* gene structures and conserved motifs confirmed the phylogenetic relationships within the *CcbHLH* TF family. Most *CcbHLH* members in each subfamily shared similar conserved motifs and gene structures, implying semblable biological functions. We found that motifs 1 and 2 occurred most frequently in the *CcbHLH* TF family, suggesting that these motifs are major components of the *CcbHLH* domain with highly conserved DNA-binding capacity [46]. In this study, the number of exons in *CcbHLH* TFs ranged from 1 to 17, which was consistent with the numbers obtained in *Panax ginseng* [37] and *Osmanthus fragrans* [43], among which 17 *CcbHLHs* had only 1 exon whereas 4 *CcbHLHs* had more than ten exons. This result suggests the possible ongoing evolution of *C. camphora bHLH* genes.

In addition, the chromosomal localization analysis suggested that fragment duplication in the *CcbHLH* gene family drove its expansion. Similar events were also found in *Liriodendron chinense* [19], *Ficus carica* L. [21], and *Pyrus bretschneideri* [17]. Furthermore, the genomic collinearity analysis of *C. camphora* and other species, as well as the *Ks* analysis of homologous genes in the collinearity block, suggested that the *C. camphora* genome had undergone three genome-wide duplication events during its evolution [29].

Mutual analysis of CcbHLH proteins can help to predict the potential functions of the *CcbHLH* TF family genes. Maize *R1*, *B1*, *Lc*, and *Sn* were the first *bHLH* TFs shown to regulate anthocyanin synthesis [47,48,49]. The homologs of maize *R* transcription factor, *TT8*, *GL3*, and *EGL3*, were then proven to function similarly in Arabidopsis [25]. In this study, we found that *CcbHLH001*, *CcbHLH022*, *CcbHLH118*, and *CcbHLH134* were orthologous to *AtbHLH42/TT8* and, therefore, they likely regulate anthocyanin synthesis in *C. camphora*.

The specificity of the expression pattern in different tissue types usually indicates the tissue-specific functions of the gene. Thus, we investigated the expression specificities of several *CcbHLHs* through transcriptomic analysis using seven different tissue types. For example, *CcbHLH005* was highly expressed in the flowers, whereas *CcbHLH042* was highly expressed in the leaves and bark. bHLH TFs from the subfamily IIIf were previously reported to regulate anthocyanin and flavonoid biosynthesis [25]. One of the subfamily members, *CcbHLH001*, was significantly more highly expressed in the stem and phloem compared to its levels in other tissues. However, the expression levels of *CcbHLH022*, *CcbHLH118*, and *CcbHLH134* did not show any tissue-specificity. Another three *CcbHLHs* previously identified to be associated with anthocyanin synthesis showed tissue-specific expression patterns as well, including *CcbHLH015* in the xylem and phloem, *CcbHLH017* in the xylem, and *CcbHLH101* in the stem and phloem. We verified the expression patterns of seven *CcbHLHs* (*CcbHLH001*, *CcbHLH022*, *CcbHLH118*, *CcbHLH134*, *CcbHLH015*, *CcbHLH017*, and *CcbHLH101*) in coloration mutant ‘Gantong 1’ at various growth stages and in different tissues by using qRT-PCR. The significantly higher expression of *CcbHLH* TFs in the bark of ‘Gantong 1’ implied their role in the anthocyanin synthesis. Taken together, the transcriptome and qRT-PCR data indicated that these *CcbHLHs* may play a crucial part in the regulation of anthocyanin biosynthesis. However, the functions of these *CcbHLHs* (*CcbHLH001*, *CcbHLH015*, *CcbHLH017*, *CcbHLH022*, *CcbHLH101*, *CcbHLH118*, and *CcbHLH134*) need to be further verified by transgenesis, the yeast one-hybrid method, and other approaches.

## 4. Materials and Methods

### 4.1. Plant Materials

Seedlings of ‘Gantong 1’, a new variety of *C. camphora*, were selected by the single plant breeding method. The young leaves of this variety are orange-red or orange, and gradually turn to yellow-green or green after maturity. The young bark is light pink with white spots and bright red after half-lignification, with obvious seasonal changes. In this study, we used three *C. camphora* ‘Gantong 1’ plants and their half-sib progeny, which were planted in the same plot at the Jiangxi Academy of Sciences (28°69′ N, 116°00′ E) in the same growth environment. All plants had a similar growth trend. ‘Gantong 1’ was the experimental group, and the half-sib progeny was the control group (Figure 7). According to the timing of the most dramatic changes in leaf and bark colors, material sampling was performed in January, April, May, July, August, and December 2021. Two-gram samples of the leaves and bark of three biological replicates of the ‘Gantong 1’ and control groups were collected, placed in a 50 mL centrifuge tube, and stored in a −80 °C ultra-low temperature refrigerator after quick freezing in liquid nitrogen.

### 4.2. Identification and Physicochemical Properties Analysis of CcbHLHs

Genome sequencing of *C. camphora* has been completed by our research group previously (GWHBGBX00000000) [29]. The Hidden Markov Model configuration file containing the bHLH domain (PF00010) was downloaded from Pfam (http://pfam.xfam.org/ (accessed on 1 October 2022)). We used online websites NCBI Batch CD-Search (https://www.ncbi.nlm.nih.gov/Structure/cdd/wrpsb.cgi (accessed on 1 October 2022)) and SMART (http://smart.embl.de/ (accessed on 1 October 2022)), to perform domain verification of the bHLH TF family protein sequences, and to exclude CcbHLHs without the bHLH conserved domain. Subsequently, we downloaded the genome data and annotation information of *Cinnamomum kanehirae* from the NCBI database (https://www.ncbi.nlm.nih.gov/data-hub/taxonomy/337451/ (accessed on 1 October 2022)) and made a preliminary identification using the same method. The annotation information of *C. camphora* was extracted from the GFF file, and the TBtools package was used to visualize it. The physicochemical properties of CcbHLH TFs were analyzed using ProtParam EXPASY online software (https://web.expasy.org/protparam/ (accessed on 3 October 2022)) [50].

### 4.3. Phylogenetic Analysis of CcbHLHs

We used the online tool MAFFT version 7 (https://mafft.cbrc.jp/alignment/software/algorithms/algorithms.html (accessed on 5 October 2022)) to perform multiple sequence alignment of the AtbHLH and CcbHLH protein sequences based on default parameters. The alignment results were uploaded to MEGA7 software, and a phylogenetic tree was constructed based on the neighbor-joining method and a bootstrap of 1000. Optional parameters were *p*-distance and pairwise deletion [51]. The evolutionary tree was drawn using the iTOL website (https://itol.embl.de/itol.cgi (accessed on 5 October 2022)).

### 4.4. Analysis of the Structure and Conserved Motifs in CcbHLHs

We used online software GSDS 2.0 (http://gsds.cbi.pku.edu.cn/ (accessed on 8 October 2022)) and MEME (https://meme-suite.org/meme/ (accessed on 8 October 2022)) to analyze the structures and conserved motifs of *CcbHLHs*, respectively, on the basis of 150 *CcbHLH* TFs cDNA sequences with corresponding genomic DNA sequences. The number of conserved motifs was set at 20. The results were visualized with TBtools [52].

### 4.5. Chromosomal Localization and Collinearity Analysis of CcbHLHs

The chromosomal localization, length, and density information were extracted from the GFF file of *C. camphora*. The genome-wide data of *A. thaliana* and *Populus trichocarpa* were downloaded from the EnsemblPlants website (http://plants.ensembl.org/index.html (accessed on 10 October 2022)), and the collinearity analysis was performed using MCScanX software [53]. The *Ka* and *Ks* values were analyzed with TBtools.

### 4.6. Protein–Protein Interaction Network Prediction

We used the online website STRING (https://cn.string-db.org/ (accessed on 15 October 2022)) to query *A. thaliana* protein sequences and predict protein interaction networks with 150 CcbHLH protein sequences as references [54].

### 4.7. Expression Levels of CcbHLHs in Different Tissues

In our previous study, we performed RNA-Seq on seven different tissues (stem, fruit, root, xylem, leaf, flower, and phloem) of *C. camphora* (‘Gantong 1’) [29]. These data were used to explore the expression pattern of *CcbHLH* TFs. The expression levels of *CcbHLHs* were estimated as kilobases per million (FPKM) reads. Tbtools software was used to visualize transcriptome FPKM data and draw gene expression heat maps based on log_2_(FPKM).

### 4.8. Analysis of Expression Levels of Candidate TFs Regulating Anthocyanin Biosynthesis

The *CcbHLHs* candidate TFs for anthocyanin biosynthesis were detected by qRT-PCR using the leaf and bark cDNA of *C. camphora* (‘Gantong 1’) and the control group, with leaf and bark as templates. Sample RNA was extracted using an RNA extraction kit (Huayueyang Biotechnology Co., Ltd., Beijing, China) and reverse-transcribed into cDNA, which was used as a template for qPCR analysis (Yeasen BioTechnologies Co., Ltd., Shanghai, China). The primers were designed using Primer Premier 5.0 software with Tm values ranging from 58 °C to 61 °C and amplified fragments ranging from 100 to 200 bp (Appendix A). ‘Gantong 1’ and the control had three biological replicates, each of which was independently repeated three times. The expression level of actin mRNA (KM086738.1) was selected as a reference, and the 2^−ΔΔCT^ method was used for calculating fold changes in gene expression levels [3]. The expression levels of *CcbHLHs* in the control sample taken in January were set as “1”.

## 5. Conclusions

This study is the first comprehensive and systematic analysis of bHLH TFs in the *C. camphora* genome. We identified 150 *CcbHLHs* that were distributed on 12 chromosomes and could be divided into 26 subfamilies. We analyzed their gene structures and conserved motifs. The collinearity analysis showed that there were 68 pairs of fragmentally duplicated genes among the 150 *CcbHLHs*. Phylogenetic analysis of *C. camphora* and *A. thaliana* revealed four candidate *CcbHLHs* (*CcbHLH001*, *CcbHLH022*, *CcbHLH118*, and *CcbHLH134*) potentially involved in anthocyanin biosynthesis in *C. camphora*. Transcriptional analysis revealed the expression pattern of *CcbHLHs* in a coloration mutant *C. camphora* ‘Gantong 1’. This study opens a new avenue for subsequent research on the functions of *CcbHLHs* in *C. camphora*.

## Figures and Tables

**Figure 1 ijms-24-03498-f001:**
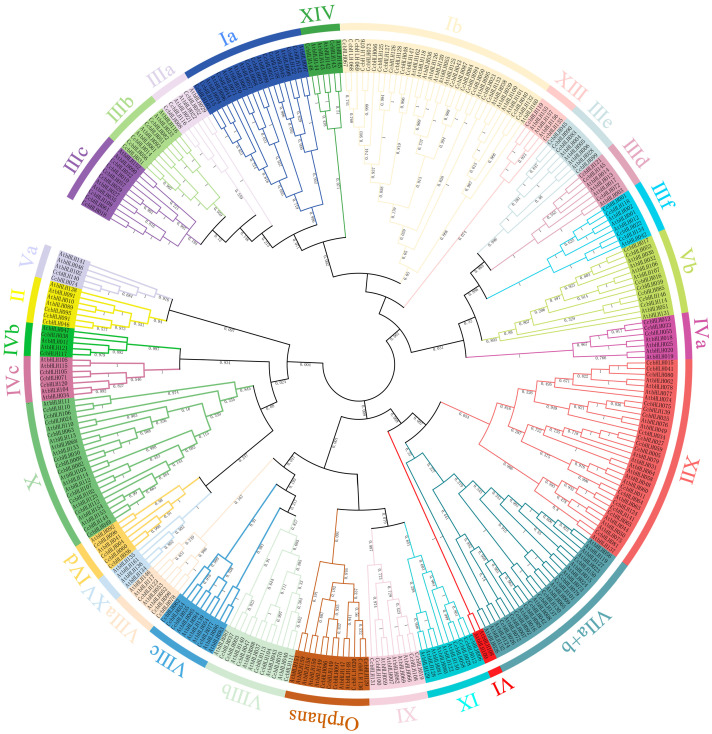
Phylogenetic relationships of bHLH proteins from *C. camphora* and *Arabidopsis*. Each subgroup is shown in a different color. The subgroup names and ID numbers are indicated in the outer circle of the phylogenetic tree.

**Figure 2 ijms-24-03498-f002:**
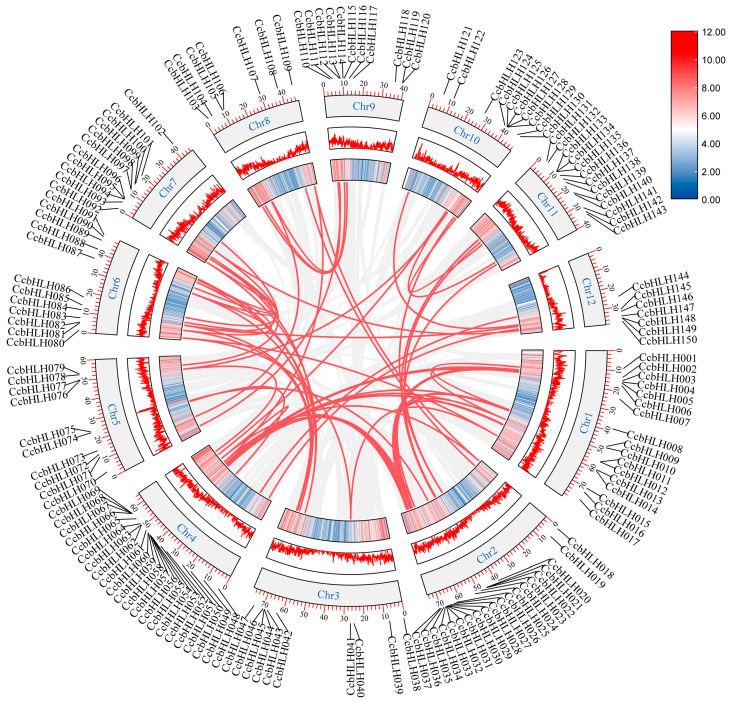
*CcbHLH* gene family segment duplication events. Segmental repeats formed *CcbHLH* genes linked by red lines.

**Figure 3 ijms-24-03498-f003:**
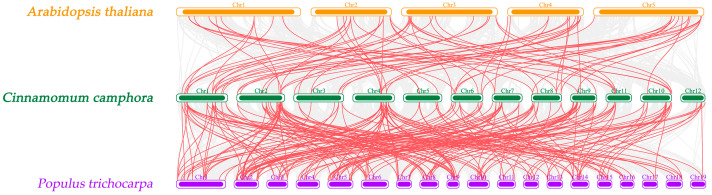
Collinearity analysis of the *bHLH* TF family in *C. camphora*, *A. thaliana*, and *P. trichocarpa*. Red lines link identified collinear genes.

**Figure 4 ijms-24-03498-f004:**
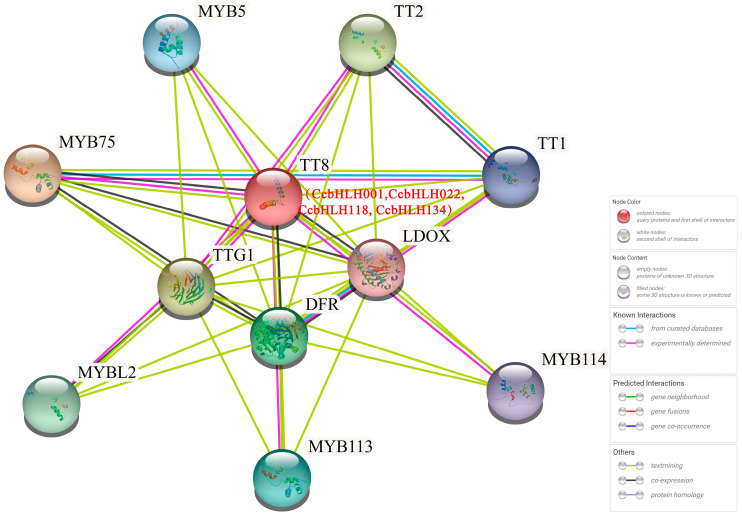
Protein interaction network prediction for CcbHLH001, CcbHLH022, CcbHLH118, and CcbHLH134 (shown in red) based on their orthologs in *Arabidopsis*.

**Figure 5 ijms-24-03498-f005:**
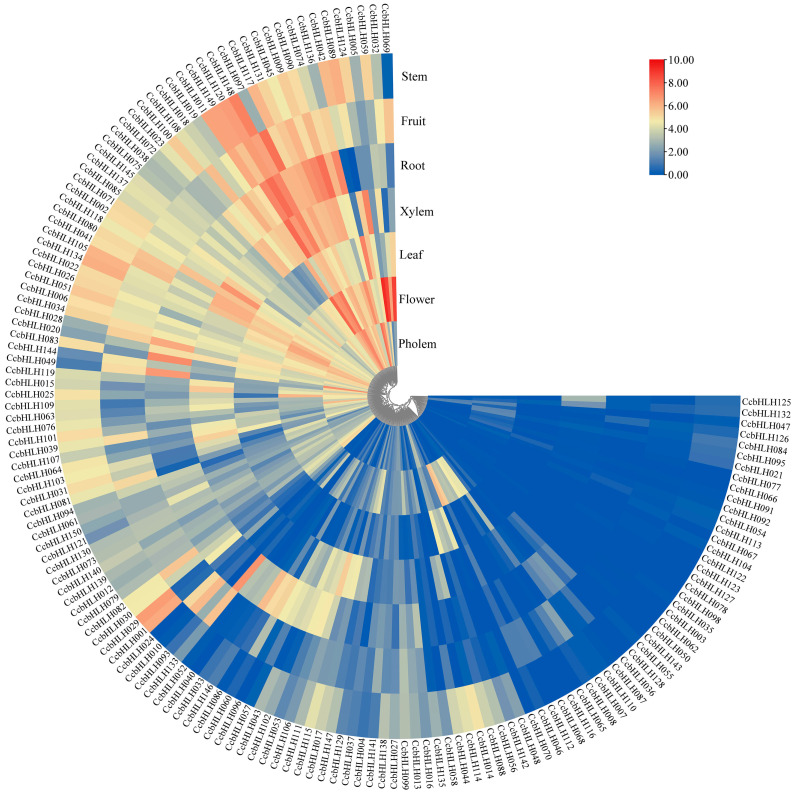
The heatmap of *CcbHLH* genes in different tissues of *C. camphora* (‘Gantong 1’). The scale bars represent the log_2_(FPKM) transformations of the FPKM values.

**Figure 6 ijms-24-03498-f006:**
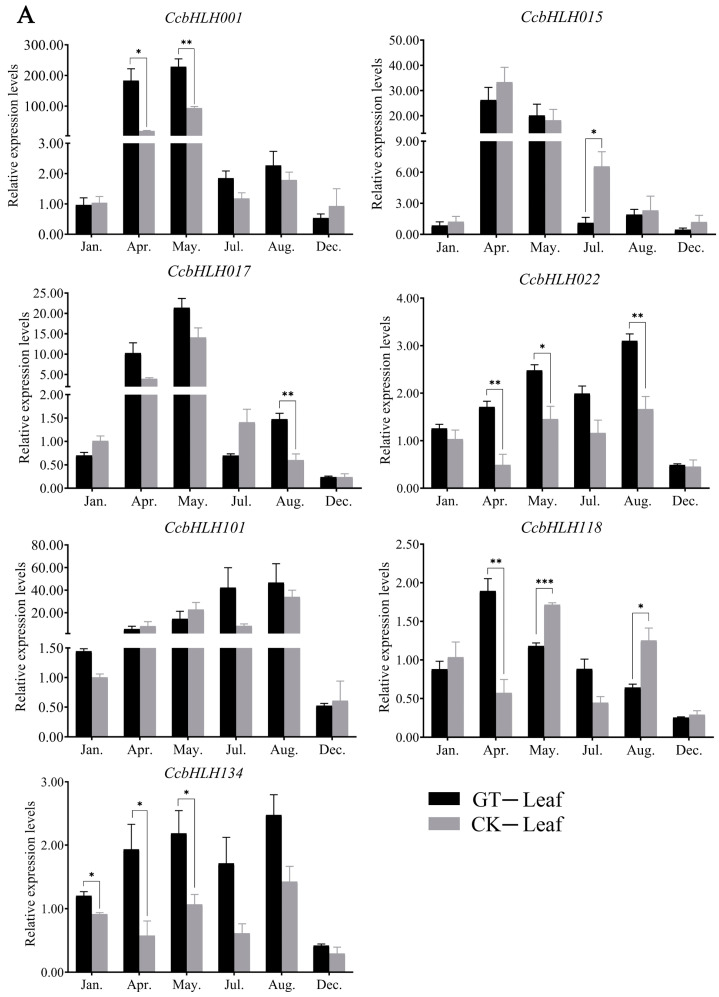
Relative expression levels of *CcbHLH* genes determined by qRT-PCR in the leaves (**A**) and bark (**B**) of *C. camphora* (‘Gantong 1’) and control plants collected year-round. Black and grey colors represent the relative expression levels in *C. camphora* ‘Gantong 1’ and the control plants, respectively. Figures with different numbers of asterisks represent significant differences on different levels. ***, *p* < 0.001; **, *p* < 0.01; *, *p* < 0.05.

**Figure 7 ijms-24-03498-f007:**
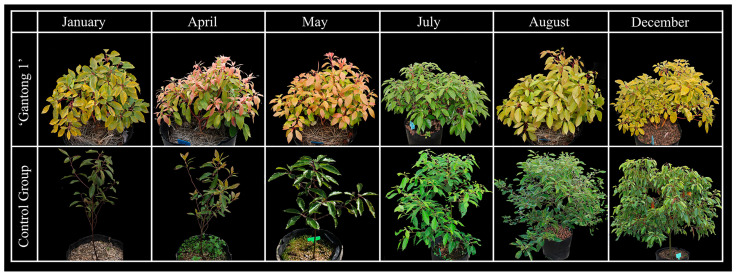
Phenotypic comparison between *C. camphora* ‘Gantong 1’ and the control group in different months.

## Data Availability

All the data are shown in the main manuscript and in the Appendix A.

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
