# Peer review of "Genome-Wide Characterization and Analysis of bHLH Transcription Factors Related to Anthocyanin Biosynthesis in Cinnamomum camphora (‘Gantong 1’)"

_ijms, 2023, doi:10.3390/ijms24043498_

Round 1

Reviewer 1 Report

In this manuscript, the bHLHs TF in C. camphora was analyzed and identified by genomic and transcriptomic analyses, and its role in C. camphora was preliminarily studied. In this work, 150 bHLH transcription factors were identified, and then the family classification and protein properties of these transcription factors were analyzed. Finally, the expression patterns of interested transcription factors were studied by RNA-seq and qRT-PCR. This work is interesting and provides a meaningful basis for studying the regulation of anthocyanin biosynthesis by CcbHLH transcription factors in C. camphora. But there are still some limitations that should be improved.

1. The correlation between "bHLH Transcription Factors" and "Anthocyanin Biosynthesis" mentioned in the title was not reflected in the results.

2. On page 2, in the MBW complex that regulates anthocyanin biosynthesis, why bHLH is selected alone in this work, and what is different from the other two transcription factors?

3. There are so many bHLH TFs have been identified in other plants, what is the novelty of identifying bHLH TFs in C. camphora?

4. On page 3, is it normal that the protein size of the same type of transcription factor varies from 10.04 kD to 136.68 kD?

5. On page 6, the calculation formula of the Ka/Ks analysis should be explained. How Ka/Ks values are associated with changes in harmful mutations? Why Ka/Ks values are less than 1 for all repeated CcbHLHs indicating that C. camphora eliminated harmful mutations through the purifying selection during the evolution? Please explain carefully.

6. On page 6, ' 60 CcbHLHs have different origins from Arabidopsis or Populus trichocarpa genes and may have specific evolution in camphor trees '. Can it be proved that these 60 bHLH TFs are specifically evolved?

7. On page 7, in the work of protein interaction network prediction of CcbHLHs, it is recommended to add yeast two-hybrid experiments to verify the prediction results.

8. On page 12, why plant materials are not collected at a uniform time interval of one year?

9. In the study of bHLH TF expression level by RNA-seq and qRT-PCR, why is there no correlation analysis of plant phenotypes to clarify the role of bHLH TF in anthocyanin biosynthesis? For example, the expression levels of bHLH TF can be comprehensively analyzed with tissue color and anthocyanin content.

Author Response

Dear Reviewers,

Thank you very much for useful comments of our manuscript. We have made careful revisions, and the detailed modification content is shown in the attachment. All changes are highlighted in the revised manuscript.

If you have any further questions, please feel free to contact me.

Best regards.

Sincerely,

Yongda Zhong

Reviewer 2 Report

In this study, authors conducted intensive in silico analyses to identify bHLH transcription factor genes from Cinnamomum camphora based on its genome sequences. Moreover, authors performed qRT-PCR for selected TF genes at different developmental stages. Overall, the quality of this manuscript is fine for publication. Introduction and results were easy to understand and nicely written. Materials and methods were written in detail. This manuscript can be acceptable after minor revision. I have few comments as follows.

The size and quality of Figure 1 should be improved.

Figure 2 is too small. I suggest the figure 2 as a supplementary figure.

Subsection 2.5 I don’t know this section is informative. It has done based on Arabidopsis protein interactions.

L240 the log2(FPKM) transformations of the RPKM values.  -> RPKM should be FPKM.

All graphs in Figure 7 should be magnified.

Subsection 4.7 Please describe about methods how to get FPKM values.    

Author Response

(The authors gave the same response as above.)

Reviewer 3 Report

Please check the attached file

Author Response

(The authors gave the same response as above.)

Reviewer 4 Report

Dear Authors

The present investigation revealed anthocyanin biosynthesis regulated by CcbHLH TFs in C. camphora, in an important tree species. The Manuscript is very well presented and discussed. I do not have any further queries or suggestions regarding present manuscript.

Good luck

Author Response

Dear Reviewers,

Thank you very much for useful comments of our manuscript.

If you have any further questions, please feel free to contact me.

Best regards.

Sincerely,

Yongda Zhong